# Material Recycling for Manufacturing Aggregates Using Melting Slag of Automobile Shredder Residues

**DOI:** 10.3390/ma16072664

**Published:** 2023-03-27

**Authors:** Soo-Jin Cho, Ha-Na Jang, Sung-Jin Cho, Young-Sam Yoon, Heung-Min Yoo

**Affiliations:** 1Resource Recirculation Research Division, National Institute of Environmental Research, Incheon 22689, Republic of Korea; 2Department of Environmental Finance, Yonsei University, Wonju 26493, Republic of Korea; 3Corporate R&D Institute in Doosan Enerbility, Changwon 51711, Republic of Korea

**Keywords:** waste, material recycling, automobile shredder residue, melting slag, environmental assessment

## Abstract

The quantity of waste from end-of-life vehicles is increasing with an increase in the number of scrapped internal combustion engine vehicles owing to international trends such as carbon neutrality and particulate matter reduction. The recycling rate must be ≥95%; however, the average recycling rate remains at approximately 89%. Therefore, the improvement of the recycling of automobile shredder residues (ASR) is gaining attention. In this study, four types of products (interlocking, clay, and lightweight swelled ceramic (LSC) bricks, and asphalt paving aggregate (APA)) were manufactured using ASR melting slag (ASRMS). Environmental performance, quality standards, and technology were evaluated to assess the recyclability of the manufactured bricks. The interlocking brick substituted melting slag for sand and stone powder as an aggregate. As melting slag content increased, absorption decreased and bending strength increased. Clay brick was manufactured by replacing kaolin and feldspar with melting slag that substituted for 20%. The quality of clay bricks mixed with over 15% melting slag was not better than standard. Asphalt paving aggregate was used to investigate the optimum condition of slag content in mixed asphalt; the mixture ratio showed that 61% broken stone of 13 mm, 6% screenings, 10% melting slag, 15% sand and 8% filler was most effective. A lightweight swelled ceramic brick was manufactured by using melting slag, front glass, and so on. Specific gravity and compressive strength ranged from 0.38 to 0.51 and from 339.7 to 373.6 N/cm^2^. ASRMS exhibited an environmental performance suitable for recycling and the manufactured bricks satisfied the quality standards. The recyclability of ASR was also assessed in terms of waste usage, conformance to quality standards, market size, and demand prediction. APA showed the best results followed by interlocking, clay, and LSC bricks.

## 1. Introduction

The number of vehicles produced worldwide over the past five years was approximately 92 million. South Korea ranks fifth in automobile production with approximately 3.5 million vehicles produced each year [1]. Internal combustion engine vehicles are being scrapped by government policies and there is an international trend to achieve carbon neutrality and particulate matter reduction. In addition, the use of electric vehicles has been incentivised. The rate of scrapping internal combustion engine vehicles and eliminating the resultant waste is expected to increase significantly [2]. Approximately 9.5 million vehicles are scrapped per year in Korea and the generated waste is approximately 103,000 tons, which is mostly composed of automobile shredder residues (ASR) [3,4]. 

Various policies have been implemented in advanced countries to efficiently recycle waste from end-of-life vehicles (ELVs). The EU’s Directive 2000/53/EC specifies that ELVs should be processed in compliance with a recycling rate of 95% and an energy recovery rate of <10%. Countries around the world have enacted legislation to introduce ELV recycling policies [5]. In Korea, there was no recycling standard for ELVs, but the Act on Resource Circulation of Electrical and Electronic Equipment and Vehicles was enacted to prepare an institutional system for ELV recycling in keeping with international environmental regulations. The act has set a mandatory recycling rate per vehicle at 95% and an energy recovery rate of <10% [6]. However, the average ELV recycling rate has only reached approximately 89% since 2015 [7]. Therefore, it is important to improve the recycling rate by recycling ASR.

ASR exhibits a high calorific value as it is composed of metals, plastics, fibres, and non-metals, such as rubber. ASR is also highly valuable as a recycling resource because it has constant waste properties with a stable generation. Yoo et al. recovered energy through the gasification of ASR [8]. Kim et al. recovered valuable metals from ASR through a melting process [9]. In this study, a method to recycle ASR as an aggregate and a road fill material was developed to improve the ELV recycling rate and establish a resource circulation economy. ASR melting slag (ASRMS) was selected as the target sample to prepare an appropriate recycling method for molten residue after energy recovery based on the results of previous studies.

The recycling of aggregate was performed by manufacturing three types of bricks using ASRMS, including interlocking, clay, and lightweight swelled ceramic (LSC) bricks. An attempt was also made to use ASRMS as asphalt paving aggregate (APA). These technologies were assessed for their applicability according to domestic laws. Therefore, analyses of the characteristics of recycled products, heavy metals, leaching, environmental performance, and quality standards were performed and conformance to the recycling standards of the Wastes Control Act were assessed.

## 2. Materials and Methods

### 2.1. Environmental Assessment of Recycling

The legal applicability of the technologies and their technical feasibilities were assessed by investigating the legal procedures and recycling standards specified in the Wastes Control Act of Korea. In the Wastes Control Act, 301 waste codes and 41 recycling types are specified. For ASR, types that can be recycled (R-3), directly recover energy (R-8), or from which energy can be recovered (R-9), and types which allow intermediate processing of waste for product manufacturing (R-10) are specified [10]. 

Korea has also implemented a system that allows the recycling of new types of recycling technologies that are not specified in the waste recycling principles provided but which comply with the requirements of an environmental assessment that evaluate their impact both with and without contact with the environmental medium. Recycling methods that have contact with an environmental medium may include recycling waste through contact with soil, groundwater, and surface water, such as road substratum materials, fill materials, and soil cover [11]. 

ASR corresponds to waste code 51-45-01 under the current law, and its recycling is allowed in types R-3, R-8, R-9, and R-10. Therefore, technologies for recycling ASR as an aggregate (R-4-2) and a road fill material (R-7) were evaluated to improve the present ELV recycling rate. Details of the classification of automobile shredder residues (ASR) and recyclable types are presented in Figure 1. 

### 2.2. Sampling and Analysis of Material Characteristics

The ASR used in this study was obtained from a shredding company in Korea, and it was composed of heavy fluff, light fluff, and glass/soil. The company compresses car bodies using a wrecker and selects materials that can be recycled. The selected materials were placed into a 660 HP pre-shredder and a 4000 HP main shredder for shredding, and light fluff (foam, dust etc.), which is a relatively light material, is separated from the materials crushed through the main shredder using a cyclone. The remaining residues are separated and recovered as iron scrap and nonferrous scrap through a separator. As for the final residue, Al, Zn, and Cu are finally discharged as heavy fluff and glass/soil through a trommel screen. Sampling was performed once per hour over three days for each ASR type. 

In this study, the ASRMS generated as residue in a study by Yoo et al., 2022, which performed energy recovery by applying gasification technology to ASR, was used. The energy recovery potential of ASR was confirmed through previous studies. In connection to this, an attempt was made to examine the possibility of material recycling along with energy recovery. There is a possibility that impurities such as organic substances may be present in the ASR. Therefore, the ASR melting slag temperature condition was sufficiently reacted at 1000 °C to control the impurties and ensure the homogeneity of the melting slag. The ASRMS was generated by a melting process in a pilot plant with a capacity of 3 ton/day. 

To evaluate the recyclability of the ASRMS, its physical and chemical properties and hazards were analysed. The density and water absorption ratios of the ASRMS were analysed in accordance with Korean Standard KS F 2503 [12]. In this instance, the water absorption ratio can be expressed as the ratio between the weight before drying and the weight after drying. It was measured using Equation (1):(1)$$a\;\left(\%\right)=\frac{m_{1}-m_{2}}{m_{1}}\times 100$$
where a: water absorption ratio (%), $m_{1}$: weight before drying (g), $m_{2}$: weight after drying (g).

In the gasification and melting system, the flow of the melt is an important factor for a reduction in the energy consumption of the melting furnace and the efficient and continuous discharge of slag. Therefore, X-ray diffraction (XRD) and X-ray fluorescence (XRF) analyses were conducted to evaluate the flow of the melt. The amorphous degree was analysed using XRD. Basicity, which affects the flow of the melt of the slag, was calculated using XRF results. Basicity is calculated using the ratio between CaO and SiO_2_ (CaO/SiO_2_), and when it is close to 1.0, the flow of the melt is considered excellent. 

In addition, various heavy metals, such as Cd, Pb, and Hg, are used in bearing cells, indicator lamps, and batteries during automobile manufacturing. Heavy metal components in the ASRMS are not likely to be eluted because they are melted at high temperatures and are composed of stable oxides. Depending on the operating conditions, heavy metals in some of the ASRMS can be eluted because they are not completely combined with the slag and exist in the form of elements or oxides. Therefore, heavy metal contents in ASR-slag and eluate were analysed using inductively coupled plasma-optical emission spectrometry (ICP-OES) (Table 1).

### 2.3. Manufacturing Bricks

Four types of products (interlocking, clay and LSC bricks, and APA) were manufactured using ASRMS, and the recyclability of each product was evaluated. The interlocking brick involves a simple paving method and it can be permanently used due to its wear resistance and lack of distortion after construction. It can also be used as a flooring material for many purposes, such as roads, pavements, and squares, due to its diversity in shape, size, and colour. It is commonly used because no special technology or heavy equipment is required for installation [13]. Clay brick is commonly used as an insulation/refractory material and a paving material due to its excellent heat storage and sound absorption capabilities. Brick that is manufactured by mixing and forming various wastes, including sludge and zeolite, as main raw materials and firing the mixture at 1150 °C or higher is referred to as LSC. LSC is used in many fields, such as architecture, civil engineering, and ship building, due to its low density and high durability. Finally, the recyclability of ASRMS as APA was evaluated. In some countries, a large quantity of ASRMS has been commercialised as an aggregate for asphalt and a material for pavement blocks [14,15]. The recycling of ASRMS as an aggregate for concrete or asphalt is based on its characteristics [16]. Each type of brick was manufactured by varying the mix proportions of raw materials and ASRMS. A schematic diagram of the process of manufacturing bricks is presented in Figure 2. In addition, the particle size distribution of sand and stone powder used in general bricks is 0.6~5.0 mm, and 1.2~10.0 mm, respectively. In order to replace ASRMS with aggregates, the size of the substitute sand was specified as 0.5~5.0 mm, according to the KS standard [17].

#### 2.3.1. Interlocking Brick

The interlocking brick was manufactured by proportioning raw materials, mixing, compression and moulding, and drying and curing. (Figure 3) The bricks were maintained at 20 °C in a separate drying room to maintain room temperature during the drying process. Three types (U and I types [220 L × 110 W × 60 H mm] and G type [260 L × 190 W × 80 H mm]) of bricks were manufactured. The mix proportions of ASRMS, sand, and stone powder were varied considering the size distribution of raw materials and brick type. The water/cement ratio was limited to <25%, in accordance with the Korean Standard (KS) 4419 [18]. The standard mix proportion for interlocking bricks was 50% sand and 50% stone powder. In this study, ASRMS was added to replace sand and stone powder and its proportions were 10, 20, 30, and 40%. U-type bricks were manufactured by reducing sand and stone powder by 5% each with the increase in the proportion of ASRMS. G-type bricks were manufactured by reducing stone powder from 40 to 10%, and the proportion of sand was fixed at 50% in accordance with the standard. In the case of I-type bricks, the proportion of stone powder followed the standard, and the proportion of sand was reduced from 40 to 30, 20, and 10%. Considering each factor, 12 variations of bricks were manufactured, that is, three types of bricks under four conditions, as summarised in Table 2. The appearance and size of bricks were examined. The bending strength and water absorption ratio were evaluated in accordance with KS F 4419.

#### 2.3.2. Clay Brick

The clay brick was manufactured through the process of mixing (hopper and roll mill), moulding, drying, and firing. KS L 4201 clay brick was manufactured using kaolin 50%, feldspar 20%, and clay 30% [19] (Figure 4). In this study, ASRMS was injected as a substitute for kaolin and feldspar, and clay bricks were manufactured by varying the proportions of ASRMS, kaolin, feldspar, and clay. The proportion of clay was fixed at 30%, whereas the proportion of ASRMS was increased to 5, 10, 15, and 20%. K-1 to K-4 bricks were manufactured by reducing the proportions of kaolin and feldspar by 2.5%. The F-1 to F-4 bricks were manufactured by reducing the proportion of feldspar by 5% without changing the proportion of kaolin. In the case of A-1 to A-4 bricks, the proportion of kaolin was reduced by 5% and that of feldspar was fixed at 20%. Table 3 summarises the conditions of clay brick manufacturing. Raw materials were pulverised to under 200 mesh, and 12 types of clay bricks were manufactured at a size of 190 L × 90 W × 60 H mm. The appearance and size of clay bricks were examined, and the compressive strength and water absorption ratio were evaluated in accordance with KS L 4201. 

#### 2.3.3. Lightweight Swelled Ceramic Brick

LSC brick was manufactured using ASRMS (20%), automotive glass (70%), and zeolite (10%) through a process of raw material grinding and mixing, moulding, firing, and foaming at 1150 °C or higher (Figure 5). The particle sizes of automotive glass, ASRMS, and zeolite were 150, 200, and 325 mesh, respectively. In the process of manufacturing bricks, 5% NaOH solution was mixed to a level of 8% of the total raw material weight to achieve a lower firing temperature. In addition, CaCO_3_ was mixed as the foaming agent at 1.5% of the total raw material weight to create pores to assist the escape of CO_2_ generated in the firing point. A total of 500 pieces of bricks were manufactured for 25 days at a rate of 20 pieces/day with a size of 350 L × 350 W ×70 H mm. The specific gravity and compressive strength of the LSC bricks were evaluated in accordance with KS L 8551 [20]. However, in the case of LSC, an evaluation of appearance and size was not performed because the bricks were prepared in accordance with the KS specifications during the foaming process. The mix proportions of materials used for manufacturing LSC bricks are provided in Table 4.

#### 2.3.4. Asphalt Paving Aggregate

The asphalt binder used in Korea is classified into ten types, according to the domestic market. The asphalt binder used in this study (AP-5) has excellent durability against plastic deformation with relatively low construction costs compared to AP-3, which is commonly used in Korea [21]. The quality of AP-5 was evaluated for the use of ASRMS as a road fill material. Table 5 summarises the quality evaluation items and criteria for APA. 

To evaluate the recyclability of ASRMS as APA, an experiment was performed by adding ASRMS into the asphalt mixture. The asphalt mixture types were compared based on Wearing Course (WC)-1, comprising of a 13 mm grinding mixture. The mixture in which 10% ASRMS was added to WC-1 was compared with that of WC-1. After 75 double-sided compactions, specimens with 0 and 10% ASRMS were cured using a mould for one day. The Marshal test was conducted to determine the optimum asphalt content (OAC) of the heated asphalt mixture and evaluate its quality. OAC influences Marshall stability, flow value, air void, saturation, and density. 

Specimens were prepared using the maximum asphalt content in the OAC range obtained through mix design, and the indirect tensile strength, recovery modulus of elasticity, and wheel tracking test were performed in accordance with ASTM D4867. The relative strength was estimated using the AASHTO TP 31-94 method by conducting the recovery modulus of an elasticity test under temperature conditions of 5, 25, and 40 °C. The properties of asphalt binder (AP-5) are shown in Table 5.

## 3. Results and Discussion

### 3.1. Characteristics of ASRMS

#### Evaluation of the Quality of ASRMS as an Aggregate

Density and water absorption ratios were analysed in accordance with the Korean Standard KS F 2503 test method, and the applicability of ASRMS as an aggregate was assessed. The results satisfied the quality standard for recycled coarse aggregate with a density of 2.24 g/cm^3^ and a water absorption ratio of 0.34% [31], indicating that it is appropriate to use the ASRMS obtained in this study as an aggregate for manufacturing bricks.

XRD analysis showed amorphous particles (Figure 6), indicating that during slow cooling, dense particles have adequate time to combine with each other, whereas low density particles float to the surface. Results of XRF analysis results show that the CaO and SiO_2_ proportions were 31.34 and 31.09%, respectively. Based on this, the flow rate affecting the fluidity of the melt was evaluated. The fluidity of the melt is an important factor in relation to equipment operating efficiency. The flow rate is considered good as the basicity (CaO/SiO_2_) gets closer to 1.0. The basicity of the ASRMS generated in this study was calculated as approximately 1.01, thereby confirming that its flow rate was excellent. Therefore, when ASR is used as feedstock, efficient equipment operation is expected, leading to continuous operation owing to the fluidity of the melt. Basicity, which affects the flow of the melts was evaluated, but pH and alkalinity were not evaluated. In addition to the operation of equipment, proportions of SiO_2_ (31.09%) and Al_2_O_3_ (19.58%) were high (approximately 50%). The materials used as aggregates in the bricks manufactured in this study, such as sand, kaolin, and feldspar, mainly consist of SiO_2_ and Al_2_O_3_; therefore, ASRMS is an appropriate substitute to these as an aggregate. The results of XRF analysis of ASR are presented in Table 6. 

### 3.2. Evaluation of Environmental Performance of Raw Materials

The heavy metals contained in ASRMS were analysed in accordance with the US EPA Method M-3050B. ASRMS had high concentrations of Fe, Al and Cr and low concentrations of Hg, As, Pb, and Cd. Heavy metal concentration is managed through the Soil Environment Conservation Act in Korea [32]. Heavy metals are not likely to be released into the environment during automobile manufacturing because they are contained in ASRMS as stable oxides during the melting process; however, some heavy metals can be eluted as they exist in the form of elements or oxides. Therefore, leaching tests of ASRMS were performed to evaluate its usefulness in recycling and landfilling. Results show that all measured potential pollutants satisfied the Korean standards. Results of heavy metal analysis suggest that recycling ASRMS in the form of bricks is an eco-friendly alternative.

However, some metal components, including Fe, had high concentrations, probably because of the compositions of heavy and light fluff, which are the main components of ASR. Although results of previous studies showed that each fluff is mainly composed of organic components, it appears that metal components were concentrated in the ASRMS through the melting process. Overall, conformance to the soil contamination standard was satisfied, but concerns over unregulated substances (e.g., Mo) remain. Therefore, ‘medium non-contact type’ recycling that does not directly come into contact with soil, such as aggregate manufacturing and recovery processes for valuable metals, are more appropriate. In particular, in the case of valuable metal recovery, the continuity of the process, waste supply stability, and economic feasibility must be secured. Therefore, this study focused on the assessment of the applicability of product manufacturing and product quality standards rather than the recovery of raw materials during material recycling. Results of the analyses of heavy metals obtained through leaching are presented in Table 7 [33].

## 4. Evaluation of Product Quality

### 4.1. Interlocking Brick

#### 4.1.1. Appearance and Size

The thickness and size of each interlocking brick was measured in accordance with the KS F 4419 standard. Ten interlocking bricks per condition were used for the measurements. Dimensional tolerances of ±2 mm for width and ±3 mm for length are prescribed by the standard. The measured bricks had dimensional tolerances of ±2 mm for width and length and −2 to −1 mm for thickness, thus satisfying the requirements of the standard. Cracks and pitting were not observed on the surface of the interlocking bricks, and they were suitable for construction. 

#### 4.1.2. Bending Strength and Water Absorption Ratio

The KS F 4419 method was used for analysis and 10 interlocking bricks per condition were tested. The bending strength was 7.8 to 8.4, 7.2 to 8.2, and 7.0 to 8.2 MPa for the U-, G-, and I-type bricks, respectively, which were above the minimum of 5 MPa specified in KS F 4419. Cement is the binder, and the bending strength of a brick increases as the cement content increases. The lower the cement content, the higher the economic feasibility. High compressive strength achieved using a small quantity of cement is an efficient brick production method. Therefore, under the same cement content, replacing sand or stone powder with ASRMS as much as possible can lead to better economic feasibility and recycling in manufacturing bricks. 

The water absorption ratio was 2.2 to 2.9, 2.4 to 3.1, and 2.4 to 3.4% for the U-, G-, and I-type bricks, which satisfied the maximum permissible water absorption ratio of 10% specified in KS F 4419 (Figure 7). Because ASRMS cannot function as a binder, the experiment was conducted with a fixed ratio of cement. When the ASRMS content increased, the water absorption ratio decreased slightly, indicating that ASRMS contributes to improving the water absorption ratio. This suggests that the water absorption ratio of ASRMS is lower than that of sand and stone powder. The long-term use of interlocking bricks is possible as the lower water absorption ratio increases durability. 

### 4.2. Clay Brick

#### 4.2.1. Appearance and Size

The quality of clay bricks was evaluated based on the KS L 4201 standard, which specifies dimensional tolerances of ±5.0, ±3.0, and ±2.5 mm for length, width, and height, respectively. In addition, the appearance must be without cracks and faults. The measured dimensional tolerances of clay bricks were 0 to +4 mm, 0 to +3 mm, and −1 to +2 mm for length, width, and height, respectively. The increase in the ASRMS content in the mix resulted in dimensions larger than that of the mould. The dimensions and appearance of all the bricks satisfied the standard. 

#### 4.2.2. Compressive Strength and Water Absorption Ratio

In order to prevent defects such as cracks that affect the strength and durability of clay brick, KS 4201 has established a standard for the compressive strength for clay bricks and evaluates their quality. Therefore, the compressive strength was evaluated along with the absorption ratio to assess the quality of the clay bricks produced in this study. The compressive strength of the clay bricks ranged from 22.8 to 46.0 MPa, which met the minimum of 22.54 MPa prescribed by the standard (Figure 8). When the contents of ASRMS in the mix were 5 and 10%, higher compressive strengths were observed and were comparable to that specified by the standard; however, when ASRMS content increased to 15 and 20%, compressive strengths were below the standard specification. It appears that the compressive strength decreased as the proportion of ASRMS increased during the firing process because the binding with other raw materials decreased. The compressive strength decreased with changes in the particle size of ASRMS. The compressive strength improved when ASRMS substituted for 10% of feldspar and kaolin rather than only feldspar. When the proportion of ASR was low and kaolin was high, the compressive strength was high. The SiO_2_/Al_2_O_3_ ratio obtained from the XRF analysis suggests that ASR had a similar ratio (1.59) to that of kaolin (1.57) compared to feldspar (2.37) [34].

The water absorption ratio ranged from 0.224 to 2.445%, which met the criterion of the standard (Figure 8). As the density of bricks increases, the water absorption ratio decreases and compressive strength increases; however, this depends on the proportions of raw materials used. 

### 4.3. Lightweight Swelled Ceramic Brick

#### Specific Gravity and Compressive Strength

The qualities of ten LSC bricks manufactured using ASRMS were evaluated in accordance with KS F 8551, which specifies a specific gravity of 0.45–0.55 and a minimum compressive strength of 294.20 N/cm^2^. The specific gravity of LSC bricks ranged from 0.379 to 0.504, indicating that some bricks exhibited values lower than the specified criterion. The average specific gravity of all the bricks was 0.453, which satisfied the standard requirement. The compressive strength of the LSC bricks was 339.7–373.6 N/cm^2^ with an average of 364.0 N/cm^2^, which satisfied the standard requirement. Compressive strength and specific gravity had the same tendency. Therefore, LSC bricks are suitable for recycling ASRMS as they are lightweight and strong (Figure 9).

### 4.4. Asphalt Paving Aggregate

#### 4.4.1. Physical Characteristics of ASRMS for Use as Aggregate

After melting and gasification, the melting slag produced after cooling has a particle size of ≤5 mm. However, the ASRMS used in this study, as shown in Figure 10, had a particle size of <1.2 mm and did not satisfy the required size distribution for use as fine aggregate. Therefore, to recycle the ASRMS used in this study as fine aggregate, a pre-treatment process is required. 

The properties of aggregate were analysed after the size distribution analysis, and the results are summarised in Table 8. The asphalt binder used in this experiment satisfied all the requirements of the standard. Therefore, the ASRMS generated in this study is suitable for use as a road filling material.

#### 4.4.2. Optimum Asphalt Content

OAC is determined through Marshall stability, flow value, porous ratio, degree of saturation, and density tests. WC-1 must satisfy a stability of 7,500 N or higher, a flow value of 20 to 40, an air void of 3 to 6%, and a degree of saturation of 65 to 80%. Results of the Marshall stability test for the specimen mixed with 0% of ASRMS show that the stability and flow value criteria were satisfied when the asphalt content ranged from 4.0 to 6.0%. In the case of air void and saturation, the criteria were met when the asphalt content ranged from 4.5 to 6.0%. Therefore, the OAC of 0% of ASRMS ranges from 4.7 to 5.5%. 

The stability and flow values of the specimen prepared by adding 10% of ASRMS met the standard criteria when the asphalt content ranged from 4.0 to 6.0%, which was similar to that of 0% ASRMS. Air void met the criterion when the asphalt content ranged from 4.5 to 5.5%, and saturation met the criterion when it ranged from 4.0 to 6.0%. Results show that the OAC of 10% ASRMS was from 4.3 to 5.0%. Compared to 0% of ASRMS, the values were reduced by approximately 0.5% when it was 10%. The factor that has the largest impact on OAC is saturation, and this tendency appears to have occurred because the saturation range of the mixture with an ASRMS content of 10% was reduced compared to the mixture with 0%. The results of the mix design with 0 and 10% ASRMS are shown in Figure 11. 

The specimen with an ASRMS content of 10% exhibited a high indirect tensile strength value (Table 9). This appears to be because the mixture manufactured by adding ASRMS is superior to typical asphalt mixtures in terms of low-temperature cracking. The results of the recovery modulus of the elasticity test on the 10% ASRMS specimen satisfied the values required for typical mixtures. Therefore, the domestic asphalt mixture exceeded the stability range based on the standard according to the Marshall test; however, results of the wheel tracking test confirmed that the plastic deformation was slightly weak. However, the cold cracking was excellent, especially the relative strength of the recovery modulus of elasticity, which was higher than those of general asphalt mixtures.

### 4.5. Final Technical Evaluation for Feasibility of Material Recycling

A technical evaluation of the bricks manufactured in this study was performed by evaluating the following four items with 100 points as a perfect score: ASR waste usage (25 points), quality standard satisfaction (25 points), market size (25 points), and demand prediction (25 points). In addition, the assessment results were derived for each evaluation item based on 25 points (excellent), 20 points (good), 15 points (moderate), and 10 points (insufficient). The assessment was performed to evaluate and predict material recycling according to the technical possibility and market size, and the environmental performance and energy consumption already mentioned above were excluded.

The ASR waste usage was evaluated by calculating the quantity of ASRMS used per manufactured brick volume. The analysis results of this study were utilised for the quality standard satisfaction item, and the market size was evaluated using the production statistics of the manufacturing industry [35]. The demand prediction item was evaluated by comparing current prices with prices in 2015 under the assumption that the demand for materials that can replace the existing aggregate will increase due to an increase in brick manufacturing costs and thus an increase in demand for bricks that use ASRMS [36]. In the evaluation results, APA, which is expected to have a large market size and high demand despite a low ASR waste usage, achieved the highest score, followed by interlocking bricks, clay bricks, and LSC bricks. In particular, interlocking bricks received a high score as they are expected to be able to treat a large quantity of waste, but they received a low score for the demand prediction item because the increase in raw material cost is not significant. Overall, clay bricks showed simple results. In the case of LSC, the lowest score was given because the market size was found to be quite small despite its excellent waste usage. Table 10 summarises the results of evaluating each brick. Overall, material recycling using ASRMS is possible; however, an efficient energy recovery process is required because significant energy consumption is expected until the discharge of the ASRMS. Further research on this issue will benefit the environment and related industries.

Currently, the world is in the process of establishing a management system to avoid simple landfills and promote recycling, and Korea needs to transition to a circular-economic society that reduces waste generation and minimizes the use of natural resources and energy. A circular economy is one that can ensure that natural resources are replaced, secured and reinvested in economic activities. If inputs are efficiently utilized within production facilities, circular resources can be secured from waste products and can reduce the amount of natural resource imports. Korea is a resource- and energy-dependent country, importing over 96% of its energy and 90% its mineral resources from abroad. Therefore, the consumption of natural resources can be reduced if valuable metals or useful resources are recovered by replacing natural raw materials or appropriately recycling them as natural aggregate substitutes or concrete additives after safe treatment. In addition, the environmental impact can be significantly reduced through the reduction of greenhouse gases and the removal of harmful substances [37].

## 5. Conclusions

This study attempted to recycle ASRMS as an aggregate for the manufacturing of four products including interlocking, clay, and lightweight swelled ceramic bricks and asphalt paving aggregate. The eco-friendly recyclability of ASRMS was evaluated through analyses of heavy metals and leaching, and appropriate quality evaluations were performed for each product. The recycling possibility of ASRMs was confirmed based on the results of this study, and the following conclusions were drawn. 

The ASRMS used in this study had a density of 2.24 g/cm^2^, a water absorption ratio of 0.34%, and a basicity of 1.01, thereby exhibiting its applicability as an aggregate. Trace quantities of Pb, Cu, As, and Cd were detected in the leaching test, but they satisfied the leaching test criteria of Korea, indicating that the ASRMS is suitable for recycling as an aggregate.ASRMS substituted for sand and stone powder as an aggregate in the manufacturing of interlocking brick. Quality evaluation of interlocking brick according to KS F 4419 in terms of appearance, size, bending strength, and water absorption ratio was satisfactory. As ASRMS content increased, absorption decreased and bending strength increased. The bending strength was ≥5.0 MPa. Therefore, a large quantity of ASR can be recycled in brick production with profits for the manufacturer due to a reduction in raw material costs.Clay brick was manufactured by replacing kaolin and feldspar with ASRMS to an extent of 20%. The quality evaluation of clay bricks according to KS L 4201 in terms of appearance, size, compressive strength, and water absorption ratio was satisfactory. However, the quality of clay bricks manufactured with >15% ASRMS was not better than that of the standard clay brick.A lightweight swelled ceramic brick was manufactured using ASRMS, zeolite, and other materials. The quality of the manufactured bricks was evaluated in accordance with KS 8551. Because all the manufactured bricks met the required criteria, manufacturing them is an appropriate recycling method for ASRMS. However, the consumption of a large quantity of ASR is unlikely for LSC because its usage is low compared to that of other bricks. It is expected that ASRMS can be utilised in many fields because it can be manufactured in various forms. Therefore, the material recycling method using ASRMS is an appropriate treatment method with many environmental benefits.To utilise ASRMS as APA for road fill materials, the particle size distribution and characteristics of the aggregate were analysed, and the OAC was obtained. The standard requirements were met for most particle size ranges, indicating that the ASRMS can be used as an aggregate. Experimental results for asphalt paving mixture with ASRMS content show that the OAC was from 4.7 to 5.5% at an ASRMS content of 0% and from 4.3 to 5.0% at an ASRMS content of 10%. As a large quantity of ASRMS can be used for asphalt manufacturing, it would be an excellent eco-friendly option.The products manufactured in this study were evaluated for recyclability in terms of waste usage, conformance to quality standards, market size, and demand prediction. Results show that APA achieved the highest rank, followed by interlocking, clay, and LSC bricks. Thus, the recyclability of ASRMS was confirmed. If this technology is continuously developed through further research, it would have positive effects on the environment and industries owing to its waste treatment and recycling potential accompanied by the lowering of the cost of raw materials.

## Figures and Tables

**Figure 1 materials-16-02664-f001:**
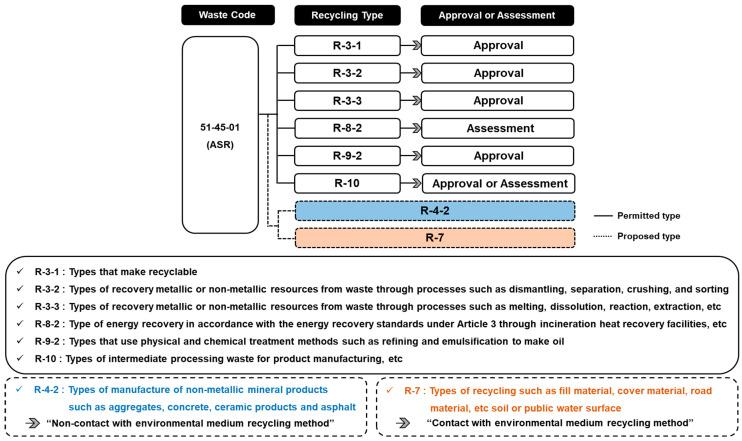
Classification of automobile shredder residues (ASR) and recyclable types.

**Figure 2 materials-16-02664-f002:**
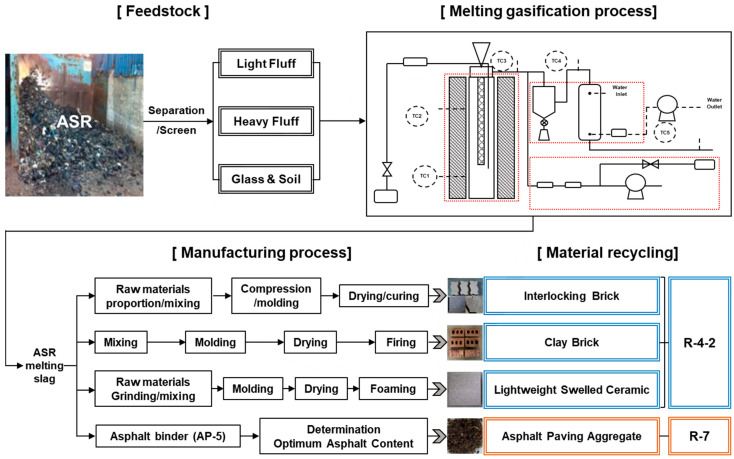
Manufacturing process of bricks.

**Figure 3 materials-16-02664-f003:**
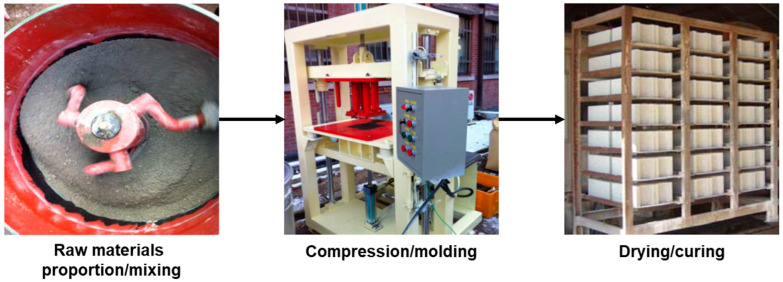
Manufacturing process of interlocking brick.

**Figure 4 materials-16-02664-f004:**
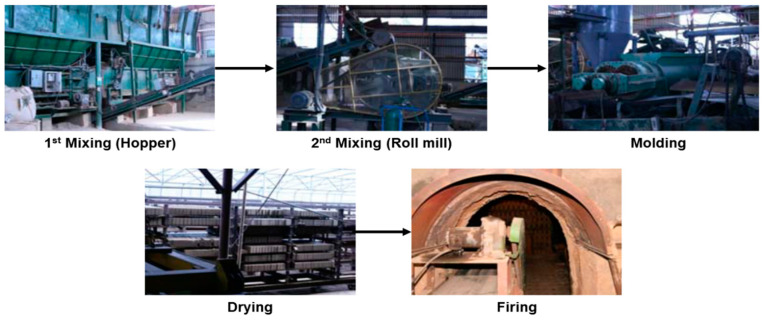
Manufacturing process of clay brick.

**Figure 5 materials-16-02664-f005:**
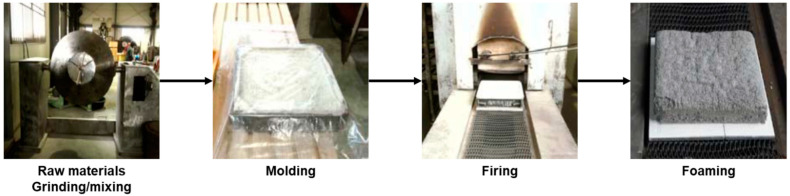
Manufacturing process of lightweight swelled ceramic brick.

**Figure 6 materials-16-02664-f006:**
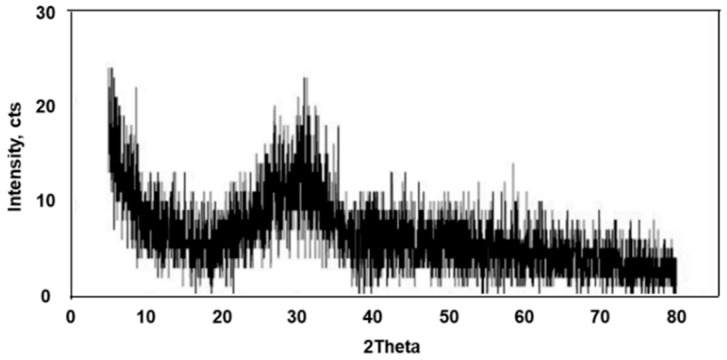
X-ray diffraction pattern of automobile shredder residues melting slag.

**Figure 7 materials-16-02664-f007:**
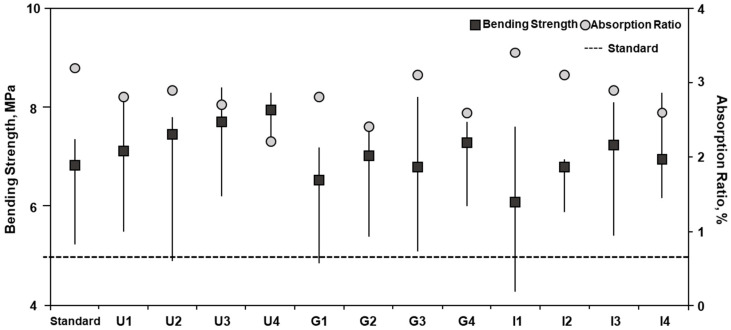
Bending strength and water absorption ratio of interlocking bricks.

**Figure 8 materials-16-02664-f008:**
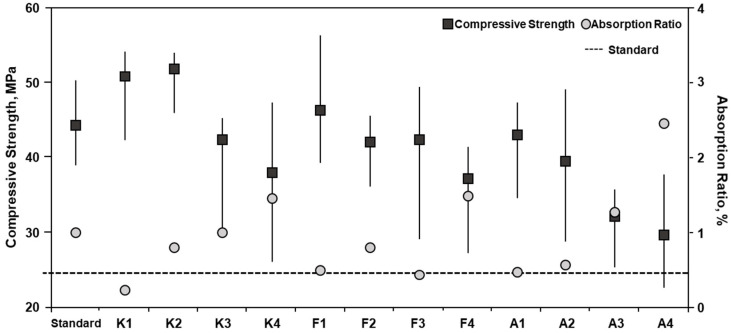
Compressive strength and water absorption ratio of clay bricks.

**Figure 9 materials-16-02664-f009:**
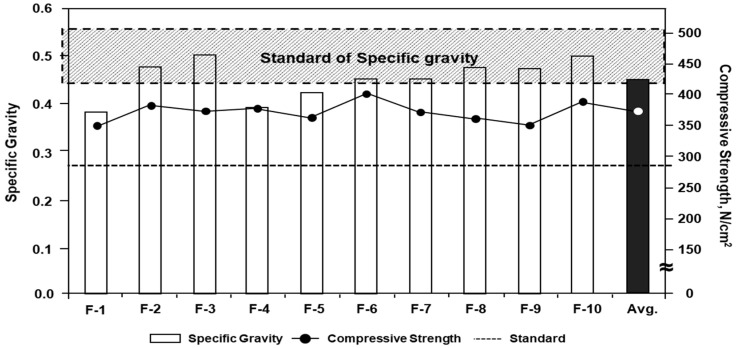
Compressive strength and specific gravity of lightweight swelled ceramic bricks.

**Figure 10 materials-16-02664-f010:**
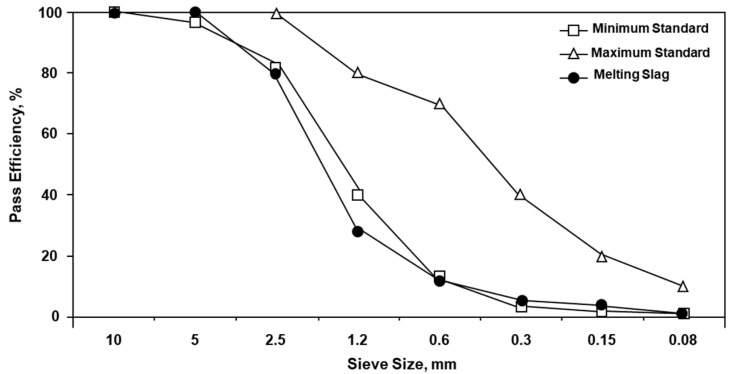
Size distribution of automobile shredder residues melting slag for use as fine aggregate.

**Figure 11 materials-16-02664-f011:**
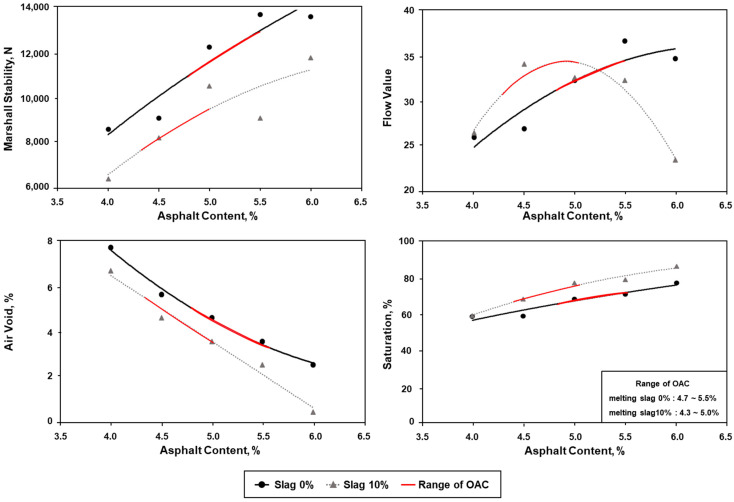
Results of asphalt paving mixtures and ranges of optimum asphalt content (OAC).

**Table 1 materials-16-02664-t001:** Analytical instruments and methods.

Material	Analysis	Instrument	Method
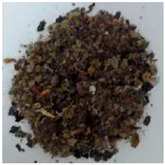	Density	Electronics densimeter MD-300S, Alfa Mirage	KS F 2503
	XRD	X-pert Pro, PNanlytical	ASTM E 82
	XRF	EDX-720, Shimadzu	ASTM D 5381
	Heavy metal	ICP-OES 720ES, Varian	US EPA Method M-3050B
	Leaching test		ES 06150. d

**Table 2 materials-16-02664-t002:** Mix proportions of cement and aggregates for manufacturing interlocking bricks.

(Unit: %)						
	Cement	Aggregate			Water/Cement	Standard (KS F 4419)
		ASRMS	Sand	Stone Powder		
Standard	25.0	0.0	50.0	50.0	20.0	Bending strength: ≥5 MPa Water absorption ratio: ≤10%
U-1		10.0	45.0	45.0		
U-2		20.0	40.0	40.0		
U-3		30.0	35.0	35.0		
U-4		40.0	30.0	30.0		
G-1		10.0	50.0	40.0		
G-2		20.0	50.0	30.0		
G-3		30.0	50.0	20.0		
G-4		40.0	50.0	10.0		
I-1		10.0	40.0	50.0		
I-2		20.0	30.0	50.0		
I-3		30.0	20.0	50.0		
I-4		40.0	10.0	50.0		

**Table 3 materials-16-02664-t003:** Mix proportions of aggregates for manufacturing clay brick.

(Unit: %)					
	ASRMS	Kaolin	Feldspar	Clay	Standard (KS L 4201)
Standard	0.0	50.0	20.0	30.0	Compressive strength : ≥24.50 MPa Water absorption ratio : ≤10%
K-1	5.0	47.5	17.5		
K-2	10.0	45.0	15.0		
K-3	15.0	42.5	12.5		
K-4	20.0	40.0	10.0		
F-1	5.0	50.0	15.0		
F-2	10.0	50.0	10.0		
F-3	15.0	50.0	5.0		
F-4	20.0	50.0	0.0		
A-1	5.0	45.0	20.0		
A-2	10.0	40.0	20.0		
A-3	15.0	35.0	20.0		
A-4	20.0	30.0	20.0		

**Table 4 materials-16-02664-t004:** Mix proportions of raw materials used for manufacturing lightweight swelled ceramic bricks.

	Material	Mixing Ratio	Size	Standard (K)
Raw material	ASRMS	20%	<150 mesh	Specific gravity: 0.45–0.55 Compressive strength : 294.20 N/cm^2^
	Automotive glass	70%	<200 mesh	
	Zeolite	10%	<325 mesh	
Foaming agent	CaCO_3_	1.5% of total raw material	<200 mesh	
Mixing solution	5% NaOH solution	8% of total raw material	-	

**Table 5 materials-16-02664-t005:** Properties of asphalt binder (AP-5).

Quality Evaluation			Standard	Method
Properties of asphalt binder (AP-5)				
Penetration (25 °C)			60–80	KS M 2252 [22]
Flash point (25 °C)			≥260	KS M 2010 [23]
Elongation (15 °C, cm)			≥100	KS M 2254 [24]
Thin firm heating	Mass change ratio (%)		≤0.6	KS M 2258 [25]
	Penetrate index residue ratio (%)		60–80	
Toluene solubility			≥99.0	KS M 2256 [26]
Softening point (°C)			44.0–52.0	KS M 2250 [27]
Penetrate index ratio after vaporisation testing (%)			≤110	KS M 2001 [28]
Elongation after heat on thin-firm			≥100	KS M 2254 [24]
Density (15 °C, g/cm^3^)			≥1.000	KS M ISO3657 [29]
Asphalt paving mixture				
Marshall test		Stability (N)	≥7500	SPS-KAI0002-F2349-5687 [30]
		Flow value	20–40	
Air void (%)			3–6	
Saturation (%)			65–80	
Indirect tensile strength test (kg_f_/cm^2^)			-	
Resilient modulus (MPs)			-	
Wheel tracking test		Dynamic stability (time/mm)	≥2500	
		Strain (mm/min)	≤0.0300	

**Table 6 materials-16-02664-t006:** Results of XRF analysis of automobile shredder residues.

(Unit: %)								
Analyte	CaO	SiO_2_	Al_2_O_3_	Fe_2_O_3_	MgO	TiO_2_	BaO	Others
	31.34	31.09	19.58	6.97	3.60	1.87	1.51	4.04

**Table 7 materials-16-02664-t007:** Results of analyses of heavy metals obtained through leaching.

Element	Concentration (mg/kg)	Leaching Test (mg/L)	
		Korean Standard	Concentration
Pb	4.75	3.0	0.31
Cu	798.77	1.0	0.30
As	1.72	1.5	0.03
Hg	0.03	0.005	N.D.
Cd	2.98	0.3	0.02
Cr^6+^	N.D.	1.5	N.D.
Cr	7000.98	-	0.04
Zn	20.73	-	0.43
Al	109,320.83	-	N.D.
Fe	178,234.25	-	0.10
Co	13.72	-	N.D.
Mn	1751.18	-	N.D.
Mo	14.94	-	N.D.
Ni	94.81	-	N.D.
Others *	-	-	N.D.

* Se, organic P, TCE (trichloroethylene), PCE (perchloroethylene), PCBs (polychlorinated biphenyl). N.D.: not detected.

**Table 8 materials-16-02664-t008:** Properties of asphalt binder.

Quality Evaluation		Standard	This Study
Penetration (25 °C)		60–80	74
Flash point (25 °C)		≥260	336
Elongation (15 °C, cm)		≥100	150
Thin firm heating	Mass change ratio (%)	≤0.6	0.04
	Penetration index residue ratio (%)	60–80	77
Toluene solubility		≥99.0	99.85
Softening point (°C)		44–52	49.50
Penetration index ratio after vaporisation testing (%)		≤110	96
Elongation after heat on thin-firm		≥100	102
Density (15 °C, g/cm^3^)		≥1.000	1.048

**Table 9 materials-16-02664-t009:** Results of quality test of asphalt paving mixture with varying automobile shredder residues melting slag (ASRMS) contents.

	Indirect Tensile Strength	Resilient Modulus (MPa) 5 °C/25 °C/40 °C	Wheel Tracking Test	
			Stability (Time/mm)	Strain (mm/min)
ASRMS 0% (Asphalt content 5.5%)	8.76	11,644/1926/1144	2916.7	0.0144
ASRMS 10% (Asphalt content 5.0%)	11.35	13,944/2986/1423	2451.4	0.0171
Standard	-	-	≥2500	≤0.0300

**Table 10 materials-16-02664-t010:** Technical evaluation for the possibility of material recycling using automobile shredder residues melting slag (ASRMS).

(Unit: point)				
Item	Interlocking Brick	Clay Brick	Lightweight Swelled Ceramic Brick	Asphalt
ASR waste usage	20	15	25	15
Quality standard satisfaction	25	15	10	25
Market size	20	15	10	25
Demand prediction	15	20	10	25
SUM	80	65	55	90
Rank	2	3	4	1

## Data Availability

Informed consent was obtained from all subjects involved in the study.

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
