# Peer review of "Material Recycling for Manufacturing Aggregates Using Melting Slag of Automobile Shredder Residues"

_materials, 2023, doi:10.3390/ma16072664_

Round 1
Reviewer 1 Report
This paper presents an experimental study of vehicle residues (ELV). The investigation is interesting as several options (4) are proposed and compared to get a final conclusion. I support publication pending the following modifications.
Corrections (line number):
20 : in terms of waste usage
79 : (interlocking, clay, LSC bricks, and APA)
Figure 2 : gasification
245 : « 0 to+ » : missing number !
References: capital first letter for journals’ names.
Supplementary
22 : complete reference for Yoo et al. (Revise presentation of the references in this file and complete them).
32-33 : the definitions of the “m” do not fit with the equation and the test done.
Table S4: what is “divide” ?
Table S5: remove all the comments.
Put a number for each equation.
General
Three (maybe four) significant digits is enough to report all the values.
Revise how to present references in the manuscript and should fit with the list reported at the end.
Author Response
I'm Soo-Jin CHO, who submitted materials-2294901 manuscript.
Thank you for your kind and consider comment.
I'll apply all of your comments.
Originally, I separated the manuscript and supplementary, but in the revision I combined the two files.
1. in terms of waste usage --> reflect
2. (interlocking, clay, LSC bricks, and APA) --> revised
3. Figure 2 : gasification --> reflect
4. « 0 to+ » : missing number ! --> 0 to 3 mm
5. References: capital first letter for journals’ names. --> revised (pp 24 ~ 28)
6. complete reference for Yoo et al. (Revise presentation of the references in this file and complete them). --> revised (pp 25)
7. the definitions of the “m” do not fit with the equation and the test done. --> revised (m1-m2)/m1*100 (pp 5)
8. Table S4: what is “divide” ? --> remove
9. Table S5: remove all the comments. --> remove
10. Put a number for each equation. --> revised
Reviewer 2 Report
The authors are asked to complete the introduction section with other type of alternatives and waste materials used in the existing literature, especially in the are of solid waste incineration fly bottom ash (MSWIBA)
- References are needed to support the results of the Characterisation of ASRMS.
- The authors should do some statistical study to ensure the homogeneity of the manufactured melting slag (ASRMS).
- Through the discussion of the results of the evaluation of product quality for interlocking block, it is recommended to analysis the relationship between bending strength and water absorption. In addition, to discuss the most important parameters to increase the bending strength is cement content or replacing sand or stone powder with ASRMS?
- It is recommended to add a methodology section that presents the experimental plan followed in this study.
It is recommended to have a figure for the relationship between compressive strength and water absorption ratio of clay brick.
It is recommended to have a figure for the relationship between compressive strength and specific gravity of lightweight swelled ceramic bricks.
- The results in this paper using of ASRMS as asphalt paving aggregate showed satisfactory results and the standard requirements of the aggregate were for most particle size ranges can be used as an aggregate. You have to compare your results with other recycling industrial materials from the optimum asphalt content (OAC) point of view. Can the use ASRMS as asphalt paving aggregate, have some advantages that can be over seen?
Author Response
I'm Soo-Jin CHO, who submitted materials-2294901 manuscript.
Thank you for your kind and consider comment.
I'll apply all of your comments.
And originally I separated the manuscript and supplementary, but in the revision I combined the two files.
- References are needed to support the results of the Characterisation of ASRMS.
—> Revised (I have added enough references for my paper.)
- The authors should do some statistical study to ensure the homogeneity of the manufactured melting slag (ASRMS). —> Revised (Added a description to pp 4)
[There is a possibility that impurities such as organic substances may be present in the ASR. Therefore, the ASR melting slag temperature condition was sufficiently reacted at 1000 to control the impurties and ensure the homogeneity of the melting slag]
- Through the discussion of the results of the evaluation of product quality for interlocking block, it is recommended to analysis the relationship between bending strength and water absorption. In addition, to discuss the most important parameters to increase the bending strength is cement content or replacing sand or stone powder with ASRMS? —> Revised (Added a description to pp 16)
[ASRMS cannot function as a binder, the experiment was conducted with a fixed ratio of cement. When the ASRMS content is increased, the water absorption ratio decreased slightly, indicating that ASRMS contributes to improving water absorption ratio]
- It is recommended to add a methodology section that presents the experimental plan followed in this study. —> Revised (Added a description to pp 4)
- It is recommended to have a figure for the relationship between compressive strength and water absorption ratio of clay brick. —> Revised (Added a description to pp 17)
[In order to prevent defects such as cracks that affects the strength and durability of clay brick, KS 4201 establishes standard for compressive strength for clay bricks and evaluates their quality. Therefore, the compressive strength was evaluated along with the absorption ratio to assess the quality of the clay bricks produced in this study ]
- It is recommended to have a figure for the relationship between compressive strength and specific gravity of lightweight swelled ceramic bricks. —> Revised (Added a description to pp 18)
[Compressive strength and specific gravity were same tendency]
- The results in this paper using of ASRMS as asphalt paving aggregate showed satisfactory results and the standard requirements of the aggregate were for most particle size ranges can be used as an aggregate. You have to compare your results with other recycling industrial materials from the optimum asphalt content (OAC) point of view. Can the use ASRMS as asphalt paving aggregate, have some advantages that can be over seen? —> Revised (Added a description to pp 24)
[If this technology is continuously developed through further research, it would be a positive effects for the environment and industries owing to its waste treatment and recycling potential accompanied by the lowering of cost of raw materials]
Reviewer 3 Report
1) More references need to be cited and reason for this study and research gap must be highlighted.
2) Many short forms are used in the manuscript it can be represented in separate table for short forms and abbreviations can be added.
3) No of samples collected and how the samples are collected needs to be explained with study area details.
4) Why researcher selected only particular heavy metal and what is the reason behind in carrying out only these particular tests.
5) How this study helps in carbon abatement and achieving SDG goals need to be explained in one paragraph.
6) If possible add some real photos of testing with Geo-tagging
7) Highlight how this study helps the stake holders/society.
7) No of references needs to be more. The given references not sufficient
8) Discussion part needs improvement.
Author Response
'm Soo-Jin CHO, who submitted materials-2294901 manuscript.
Thank you for your kind and consider comment.
I'll apply all of your comments.
And originally I separated the manuscript and supplementary, but in the revision I combined the two files.
1) More references need to be cited and reason for this study and research gap must be highlighted.
—> Revised (I have added enough references for my paper. pp 24-28)
2) Many short forms are used in the manuscript it can be represented in separate table for short forms and abbreviations can be added.
—> Revised (I have added abbreviations for my paper. pp 2)
3) No of samples collected and how the samples are collected needs to be explained with study area details.
—> Revised (Added a description to pp 4-5)
4) Why researcher selected only particular heavy metal and what is the reason behind in carrying out only these particular tests.
—> Revised (Added a description to pp 15)
[For content, we analyzed a total of 19 substances that are expected to be present with high probability. For leaching test, we mainly analyzed substances specified in the KOREA STANDARD.]
5) How this study helps in carbon abatement and achieving SDG goals need to be explained in one paragraph.
—> Revised (Added a description to pp 22)
[Korea is a resources and energy dependent country, importing over 96% of its energy and 90% its mineral resources from abroad. Therefore, consumption of natural resources can be reduced if valuable metals or useful resources are recovered by replacing natural raw materials or appropriately recycled as natural aggregate substitutes or concrete additives after safe treatment. In addition, the environmental impact can be significantly reduced through the reduction of greenhouse gases and the removal of harmful substances]
6) If possible add some real photos of testing with Geo-tagging
—> Revised (Added a figures. pp 7, 9, 10)
7) Highlight how this study helps the stake holders/society.
—> Revised (Added a description to pp 22)
[Currently, the world is in the process of establishing a management system to avoid simple landfills and promote recycling, and Korea needs to transition to a circular-economic society that reduces waste generation and minimizes the use of natural resources and energy. Circular economic that can replace natural resources are secured and reinvested in economic activities, if the inputs are efficiently utilized within the production facility, circular resources can be secured from waste, reducing the amount of natural resource imports]
7) No of references needs to be more. The given references not sufficient
—> Revised (I have added enough references for my paper.)

Reviewer 4 Report
The topic of this paper is interesting and contribute knowledge to the field of recycle material. However, there are some corrections needed to improve it. Discussion for each products should be elaborated more significance including adding more tables of results and data explanation.Others comments shall be found in the paper attached. Paper focusing on too many materials and explanation for each materials are too brief. Therefore, suggest to elaborate extensively.Reference format shall be checked and add more references.

Author Response
'm Soo-Jin CHO, who submitted materials-2294901 manuscript.
Thank you for your kind and consider comment.
I'll apply all of your comments.
And originally I separated the manuscript and supplementary, but in the revision I combined the two files.
Take a look at my entire paper, revising all of your revision comments!

Reviewer 5 Report
1. In section 2.1, I suggest authors to discuss the existence of possible impurities in ASR and is any method required to remove the impurities before using in any applications.
2. In section 3.1.1, I recommend authors to mention the size of ASRMS aggregate used in the present study and reason for the selection of particular size.
3. In section 4.2.2, Authors must clarify, is there any relation between compressive strength and method adopted for pressing the bricks?
4. Is any alkalinity test conducted in clay bricks? I suggest authors to clarify.
5. In many sentences authors come to explain the technical terms with abbreviations. The explanation word for those abbreviations can be in title case.
6. Authors must check for reference part mentioned is as per journal template.
Author Response
I'm Soo-Jin CHO, who submitted materials-2294901 manuscript.
Thank you for your kind and consider comment.
I'll apply all of your comments.
And originally I separated the manuscript and supplementary, but in the revision I combined the two files.
1. In section 2.1, I suggest authors to discuss the existence of possible impurities in ASR and is any method required to remove the impurities before using in any applications. —> Revised (Added a description to pp 4)
[Therefore, the ASR melting slag temperature condition was sufficiently reacted at 1000 to control the impurties and ensure the homogeneity of the melting slag]
2. In section 3.1.1, I recommend authors to mention the size of ASRMS aggregate used in the present study and reason for the selection of particular size.
—> Revised (Added a description to pp 6)
[In addition, the particle size distribution of sand and stone powder used in general bricks is 0.6 ~ 5.0 mm, 1.2 ~ 10.0 mm respectively. In order to replace ASRMS with aggregates, the size of the substitute sand was specified to 0.5 ~ 5.0 mm according to the KS standard]
3. In section 4.2.2, Authors must clarify, is there any relation between compressive strength and method adopted for pressing the bricks? —> Revised (Added a description to pp 17)
[In order to prevent defects such as cracks that affects the strength and durability of clay brick, KS 4201 establishes standard for compressive strength for clay bricks and evaluates their quality. Therefore, the compressive strength was evaluated along with the absorption ratio to assess the quality of the clay bricks produced in this study]
4. Is any alkalinity test conducted in clay bricks? I suggest authors to clarify.
—> Revised (Added a description to pp 13)
[Basicity, which affects flow of the melts was evaluated, pH and alkalinity were not evaluated]
5. In many sentences authors come to explain the technical terms with abbreviations. The explanation word for those abbreviations can be in title case.
—> Revised (I have added abbreviations for my paper. pp 2)
6. Authors must check for reference part mentioned is as per journal template.
—> Revised (I have revised references for my paper. pp 24-28)
Round 2
Reviewer 3 Report
Authors have addressed the comments. It can be accepted .
Reviewer 4 Report
Paper revised as per requested. It shall be accepted for publication as it.